# The impact of the three-level digital divide on the mental health of rural residents: A study from China

Yi Ding🔗, Yunhui Ai🔗*

Vocational and Technical College, Inner Mongolia Agricultural University, Baotou, Inner Mongolia Autonomous Region, China

* 18104850327@163.com

## Abstract

This study adopts the theoretical framework of "digital access divide—digital use divide—digital utility divide" to systematically investigate the differential impacts of different levels of the digital divide on the mental health of rural residents in China and its influencing mechanisms. Based on panel data from the China Family Panel Studies (CFPS) during 2020–2022, the study analyzes the internet access, usage, and perceived importance of rural residents in China, i.e., the situations of digital access divide, digital use divide, and digital utility divide they encounter, to examine the effects of these three levels of the digital divide on the mental health of rural residents.The findings are as follows: First, the digital access divide directly leads to a decline in mental health levels, and the digital use divide and digital utility divide, characterized by insufficient usage capabilities and improper usage behaviors, further exacerbate psychological damage. Second, different social groups exhibit heterogeneous results under the influence of the digital divide, with factors such as educational level and regional differences moderating the intensity of the digital divide's impact. Third, there are hierarchical differences in the mechanisms of influence: The digital access divide reduces the mental health level of farmers by lowering their self-assessed sense of fairness. The digital use divide reduces the mental health level of farmers by lowering their self-assessed social class. The impact of the digital utility divide on the mental health of rural residents is achieved by reducing both self-assessed social class and self-assessed economic status. This study provides a new analytical perspective for understanding the mental health issues of rural residents in the context of digital technology popularization and has reference value for formulating precise digital inclusion policies.

**Data availability statement:** All relevant data are within the paper and its Supporting information files.

**Funding:** The author(s) received no specific funding for this work.

**Competing interests:** The authors have declared that no competing interests exist.

## 1. Introduction

The comprehensive innovation of digital information and communication technologies (ICTs) has transcended the scope of information sharing, bringing new digital opportunities and dividends to individuals and society. Since the proposal of the "Digital China" development strategy in 2015, its goal has been to promote the all-round digital transformation of social production, business, healthcare, and other fields. After nearly a decade of advancement, China has made significant progress in internet infrastructure construction, and the digital access divide has been markedly reduced. According to the 55th "Statistical Report on China Internet Development" released by the China Internet Network Information Center (CNNIC), as of January 2025, the number of internet users in China is close to 1.1 billion, and the internet penetration rate has reached 78.6% [1].

However, during the rapid development of internet technology, not all residents can enjoy equal opportunities for digital information access, and this inequality in information access opportunities is an important manifestation of the digital divide. The concept of the "digital divide" was originally defined as differences among individuals regarding the acquisition of internet material resources. Essentially, it refers to whether individuals can properly access the internet or use internet-enabled devices. This phenomenon is also known as the "digital access divide" or "first-level digital divide" [2]. With the development of technology, the concept of the digital divide has been extended on the original basis. For individuals with converging internet access opportunities, they may also have varying levels of usage due to a lack of knowledge and skills in using digital resources, and this difference is referred to as the "digital use divide" or "second-level digital divide" [3]. In recent years, some researchers, based on the concept of the second-level digital divide and using the perceived importance of internet use by individuals as a criterion, have proposed the concept of a "third-level digital divide." The third-level digital divide refers to the gap in individuals' perceived importance of internet use, which can significantly impact residents' lives, work, and welfare levels [4].

In the context of the digital divide, rural residents constitute a distinct and deeply affected group. Although often collectively referred to as farmers, this population in fact displays significant occupational heterogeneity, encompassing not only those engaged in agricultural production but also local non-agricultural workers, return-migrant entrepreneurs, retirees, and other long-term inhabitants of rural China—rather than being restricted to farmers in the traditional sense. Consequently, the present study is situated within this demographically diverse rural population rather than being limited to individuals whose primary activity is farming.

In the context of the digital divide, rural residents are a unique and deeply affected group. According to the 2021 Seventh National Population Census data in China, the rural population is 509.79 million, accounting for 36.11% of the total population. Given the significant proportion of rural residents in the total population, their quality of life and mental health not only affect their personal development but also have a significant impact on the stability and harmony of society as a whole [5]. However, due to the existence of the digital divide, the farming community faces significant challenges

in information access and digital skills, hindering their ability to enjoy the benefits brought by the development of digital technology. In the long term, this will seriously impede their development in areas such as work, entertainment, and social interaction.

In the context of the vigorous development of the "Digital China" strategy, addressing the digital divide and mental health issues of Chinese farmers is of great significance for maintaining national stability and public welfare. According to the self-regulation theory, people are not only influenced by the external environment but can also affect their behavior and thinking by regulating their mental health [6]. Therefore, it is even more necessary to study the relationship between the digital divide and the mental health of farmers. This paper introduces a two-way fixed-effects model to deeply explore the impact of the digital divide on the mental health of rural residents in China and further investigates the mechanisms of influence. Thus, in the measurement of the digital divide, this study adopts the research logic of "digital access divide—digital use divide—digital utility divide" and deeply explores the different impacts of each level of the digital divide on mental health. In addition, this paper also focuses on the moderating role of intermediary variables, including self-assessed social class perception, self-assessed economic status, and self-assessed sense of fairness, in the relationship between the digital divide and mental health.

The main contributions of this study are as follows:

First, in terms of the research status of the digital divide and the mental health level of rural residents. The digital divide and the mental health level of rural residents are two fields with weak intersection, and the current understanding of their relationship is still limited. Existing research mostly focuses on the impact of the "digital access divide" on residents' mental health, but the relevant findings are contradictory [7,8], which makes it challenging to conduct a systematic analysis of the impact of the digital divide on the mental health level of rural residents. This paper systematically explores the impact of different levels of the digital divide on the mental health level of rural residents in China based on the research logic of "digital access divide—digital use divide—digital utility divide."

Second, in terms of the interconnection and research difficulties of the three levels of the digital divide. The three levels of the digital divide, namely, the access divide, the use divide, and the utility divide, are interrelated and influence each other through chain effects [9]. However, due to limitations in data and other factors, existing research finds it difficult to systematically analyze the manifestations of different levels of the digital divide at the individual level. This study rigorously measures the different levels of the digital divide and elaborates on its diverse forms of expression, providing more operational ideas and frameworks for future research.

Third, in terms of the mechanisms of the digital divide's impact on the mental health of rural residents. Existing research generally lacks an in-depth exploration of the mechanisms through which the digital divide affects the mental health level of rural residents [10]. This study focuses on examining the impact of the digital divide on the mental health of rural residents through factors such as self-assessed social class perception, self-assessed economic status perception, and self-assessed sense of fairness, and further tests the robustness of the relevant conclusions at the three levels of the digital divide.

Fourth, in terms of the regional limitations and deeper significance of digital divide research. Existing literature on the impact of the digital divide on residents' mental health is mostly concentrated in developed regions, with relatively few case studies in developing countries [11]. As a typical developing country, China's digital divide issues are unique and representative but have not yet been systematically studied and comparatively analyzed. This study focuses on rural areas in China, filling the gap in existing research in developing countries and providing new perspectives and empirical evidence for global digital divide research.

## 2. Theoretical mechanisms

### 2.1. The impact of the digital divide on the mental health of farmers

**2.1.1. Measurement and influencing factors of farmers' mental health.** In contemporary society, with the continuous development of socio-economic levels and the intensification of competitive pressures, mental health issues

among residents, characterized by psychological imbalance and emotional disorders, have shown a gradually deepening trend and have become problems that society must address [12,13]. Consequently, in recent years, numerous scholars have focused their research on the mechanisms through which the digital divide affects residents' mental health.

Empirical research on residents' mental health has gradually increased internationally. In measuring mental health levels, existing literature primarily employs two approaches: single-dimensional and multi-dimensional measurement. Single-dimensional measurement typically relies on a single indicator, such as depression or health, for example, by assessing mental health levels through individuals' self-evaluations of subjective well-being. A commonly used method is the modified GSS 1972 measurement, which asks, "What is your level of happiness recently: 1 Very happy, 2 Moderately happy, 3 Unhappy." Additionally, single-question Likert scales are also widely applied [14].

Multi-dimensional measurement involves comprehensively evaluating mental health levels using multiple factors. In research, methods such as factor analysis, structural equation modeling, and polyhedral measurement are commonly employed. Factor analysis is advantageous in revealing the intrinsic relationships between independent and dependent variables [14]. Structural equation modeling can measure the relationships between latent variables and observed variables [15]. Polyhedral measurement enhances the objectivity and robustness of data by utilizing geometric models to display data distribution [16].

Both measurement approaches have their advantages and limitations and are widely used in research. Single-dimensional measurement is advantageous due to its low data requirements and simplicity. However, its limitations lie in the restricted accuracy and comprehensiveness of the research. Multi-dimensional measurement, on the other hand, yields more accurate conclusions and accounts for the potential impacts of influencing factors, but its limitations include high data requirements, complex procedures, and difficulties in cross-national comparisons.

In recent years, numerous scholars have begun to focus on potential factors influencing residents' mental health levels. Current research typically unfolds at three levels: micro-individual, meso-family, and macro-societal. At the micro-individual level, studies primarily focus on individual characteristics such as gender, education level, household registration type, personal income, and personality traits [17–19]. At the meso-family level, research pays more attention to family factors, including the intimacy of family relationships, family property conditions, number of children, and marital satisfaction [20,21]. At the macro-societal level, studies focus on the macro-social background of the subjects, such as urbanization levels, aging rates, internet penetration rates, medical conditions, and social welfare [22–24].

Farmers, as a significant group constituting over one-third of China's population, face unique digital divide issues. First, the digital divide among farmers is multi-dimensional, not only reflected in insufficient physical access to devices (e.g., low smartphone and internet coverage rates) but also involving differences in usage capabilities and knowledge acquisition. For instance, Zhu et al. pointed out that farmers in economically underdeveloped regions, due to weak infrastructure and unequal distribution of technical resources, struggle to effectively participate in the digitalization process, resulting in significant gaps in information access and social resource utilization compared to urban residents [25]. Second, the generally low education levels of farmers directly limit their digital literacy. Research indicates that educational attainment is positively correlated with digital skills, and the scarcity of educational resources in rural areas further exacerbates the disadvantaged position of farmers in digital technology applications, making it difficult for them to adapt to modern digital scenarios such as online services and telemedicine. Chai's research further highlights that the "digital utility divide" is particularly prominent among farmers. Due to insufficient information discernment capabilities, they are more susceptible to online fraud and false information, thereby increasing psychological anxiety and insecurity. This dual dilemma of technological adaptation lag and psychological stress not only marginalizes farmers materially in the digitalization process but also exposes them to mental health risks [26].

Moreover, although existing research has predominantly focused on agricultural populations in rural areas, mounting evidence indicates that non-agricultural rural residents—such as return-migrant entrepreneurs, local non-farm workers, and retirees—also face a twofold disadvantage stemming from both digital access divides and insufficient digital-use

 

capabilities. For instance, Wang found that rural residents' access to health care and mental-health treatment is constrained by limitations in internet access and use [27]. Lee further observed that among older rural residents the effects of internet access, use, and perceived utility are more severe than among their non-rural counterparts and vary across occupations. These studies suggest that the impact of the digital divide may exhibit intra-rural heterogeneity contingent upon occupational status and social identity [28].

However, despite existing research describing and explaining the impact of the digital divide on residents' mental health levels, studies on the impact of the digital divide on the mental health of residents in rural China remain relatively limited. Given the uniqueness of China's farmer population, research on the impact of the three levels of the digital divide on the mental health of rural residents is of significant importance. It not only provides references for policymakers in bridging the digital divide among rural residents in China but also offers suggestions for improving the digital divide levels of farmers in other developing countries.

**2.1.2. The concept and research of the digital divide.** The concept of the digital divide can be traced back to the 1999 report "Falling Through the Net: Defining the Digital Divide" released by the National Telecommunications and Information Administration (NTIA) of the United States. The report defined the digital divide as the gap between those who possess the tools of the information age and those who do not. Since then, with the rapid development of information technology, the connotation of the digital divide has deepened, and research has gradually expanded from the level of access to the levels of use and utility [29].

As research has deepened, the concept of the digital divide has been continuously refined. The classification framework of "digital access divide—digital use divide—digital utility divide" is now widely accepted in academia. Initially, research on the digital divide focused on the access divide, that is, the inequality in internet access and the acquisition of digital information resources among different countries, regions, or social groups. With the improvement of internet development levels, researchers found that even within the same country or region, there are significant individual differences in internet usage extent and frequency. This type of usage difference, which is based on access levels, is referred to as the use divide [30].

In recent years, with the continuous improvement of internet penetration rates, research on the digital divide has gradually extended to the third level, namely the "utility divide." This level focuses on the long-term impacts brought by differences in internet access and usage. Research has found that the long-term effects of internet access and usage are far more profound than the differences in access and usage themselves [31]. For example, differences in internet usage levels not only affect individuals' digital skills and information literacy but may also further exacerbate social inequalities. These differences manifest in people's perceptions, values, and behavioral norms, and may even impact personal career development and social participation.

Moreover, research indicates that the three levels of the digital divide—access, use, and utility—are not independent but interact through chain effects, collectively exerting multi-dimensional impacts on individuals [32]. Objectively, the digital divide significantly hinders the improvement of social welfare, employment opportunities, income levels, asset accumulation, and educational levels, thereby further intensifying social inequality and social stratification. For instance, the digital divide has a significant negative impact on the non-agricultural employment quality of rural residents, particularly among low-skilled groups [33]. Subjectively, the digital divide also negatively affects individuals' mental health, quality of life, and physical health. Studies have found that the digital divide reduces subjective well-being and increases the frequency of psychological depression and negative emotions [34]. These effects are particularly pronounced among vulnerable groups such as rural residents, individuals with lower educational levels, and the elderly, who find it harder to benefit from digital transformation and are more sensitive to the impacts of the digital divide [35].

**2.1.3. The impact of the digital divide on residents' mental health.** Currently, there is limited research exploring the intersection between the digital divide and residents' mental health. These topics are often studied independently within their respective fields, resulting in the formation of separate knowledge systems. In existing studies, researchers typically

use indicators such as internet access rates, usage patterns, and the importance of internet usage to measure the levels of the digital divide. The impact of the internet on residents' health is primarily investigated through two pathways: one is the direct pathway, focusing on digital health consultations, smart medical services, and health monitoring applications; the other is the indirect pathway, focusing on mental health and happiness levels. Existing research indicates a positive correlation between internet usage frequency and residents' mental health levels, and the main reasons are as follows:

First, the "stimulation hypothesis" posits that internet usage, particularly the use of new media, can effectively increase the time and quality of social interactions among residents. Additionally, it indirectly promotes the fulfillment of basic psychological needs such as a sense of belonging and a sense of value, thereby enhancing mental health levels [36].

Second, the widespread use of the internet provides residents with a new platform for expressing emotions and personal opinions, which indirectly enhances their sense of achievement and fairness, contributing to improved mental health [37].

Third, the extensive use of the internet has, to some extent, contributed to increased regional employment levels and growth in per capita disposable income, which indirectly enhances residents' mental health levels [38,39].

Beyond these findings, related literature also highlights that inappropriate internet usage may have adverse effects on residents' mental health. For example, excessive internet usage may intensify individuals' feelings of social comparison, leading to a decline in mental health levels. Valkenburg synthesized existing literature from 2019 to 2021 and noted that the proliferation of social media can heighten emotions such as envy and social comparison between different classes, which negatively impacts mental health [40]. Additionally, Carraturo conducted a literature review examining the relationship between social comparison on the internet and psychological depression, finding that the use of social media platforms like Facebook can increase users' tendencies toward social comparison, leading to adverse outcomes such as declining mental health levels and increased depressive moods [41].

These studies collectively demonstrate that the impact of internet usage on mental health is not unidirectional but rather depends on a combination of factors, including usage motivation, usage patterns, and usage frequency. Therefore, understanding the complex impact of internet usage on mental health requires a comprehensive consideration of individuals' usage habits and their social environment.

## 2.2. Research hypotheses

As previously discussed, existing research on the impact of internet use on residents' mental health has yielded relatively divergent conclusions. Some scholars argue that internet application promotes residents' mental health, while others hold the opposite view [42,43]. This study posits that due to China's relatively late start in internet development and significant urban-rural disparities, the depth and breadth of internet technology application among rural residents in China remain to be improved. Therefore, we hypothesize that higher levels of internet use are conducive to improving the mental health of rural residents. Conversely, a deepening digital divide would negatively affect the mental health of rural residents in China. Specifically, the digital access divide restricts rural residents' channels for information search, hindering their access to information beneficial for improving mental health. The digital use divide exacerbates differences in internet usage capabilities among individuals, preventing rural residents from leveraging the low-cost and high-efficiency advantages of internet technology, thus negatively impacting mental health. Additionally, excessive internet use may also adversely affect the mental health of rural residents. Based on this, we propose Hypothesis H1:

**H1: All levels of the digital divide, including the digital access divide, digital use divide, and digital utility divide, will have adverse effects on the mental health of rural residents.**

Self-Assessed Social Class Perception and Self-Assessed Economic Status Mechanisms.The most direct impact of widespread internet technology is that it significantly broadens residents' social networks, enabling them to learn more about the economic and knowledge levels of individuals from different social classes, thereby prompting them to reassess their own social class. However, existing research has not reached a consensus on this effect. Some

researchers adopt a positive stance, arguing that the internet provides an open and information-sharing platform for rural residents [44], where individuals often compare themselves with groups of lower social or economic status, which may enhance their self-assessed social class and economic status. Conversely, others argue that individuals tend to compare themselves with higher-status groups, potentially lowering their self-assessed social class or economic status [45].

Additionally, a series of studies have shown a strong association between mental health and social class. For example, Xu suggest that individuals of higher social class maintain better mental health because higher social status enhances their sense of social recognition, driving improvements in mental health [46]. Similarly, an improved self-assessed economic status also promotes better mental health. Therefore, we propose Hypotheses H2 and H3:

**H2: All levels of the digital divide reduce the mental health of rural residents by lowering their self-assessed social class perception.**

**H3: Different levels of the digital divide reduce the mental health of rural residents by decreasing their self-assessed economic status perception.**

Self-Assessed Sense of Fairness Mechanism.Scholarly consensus on how internet use influences residents' perception of fairness remains elusive. Some scholars argue that internet use may negatively affect individuals' sense of fairness. For instance, research has found that media in developed countries often propagate information about their own economic prosperity, which can confuse residents of developing countries and deepen their perception of social inequality and uneven development [47,48]. Conversely, others contend that internet penetration provides a platform for diverse viewpoints, conducive to enhancing individuals' understanding of fairness and democracy Building on the arguments of Chen, Guo, and Benvenuti, who contend that governments can raise rural residents' perceived sense of fairness by rigorously censoring media content, filtering negative online information, and actively guiding public opinion [49–51], the present study posits that the persistence of the digital divide is likely to exert a deleterious effect on rural residents' sense of fairness.

Furthermore, research has found that social injustice increases residents' depression levels or reduces their happiness, ultimately harming their mental health [52]. Thus, we propose Hypothesis H4:

**H4: The digital divide reduces the mental health of rural residents by lowering their self-assessed sense of fairness.**

Fig 1 outlines the research logic of this paper. As shown in Fig 1, this paper aims to investigate the impact of the digital divide at different levels on the mental health levels of rural residents. The digital divide primarily includes the digital access divide, the digital use divide, and the digital utility divide. On this basis, this paper seeks to explore whether self-assessed social class perception, self-assessed economic status, and self-assessed sense of fairness mediate the adverse effects of each level of the digital divide on rural residents.

## 3. Data and variables

### 3.1. Sources of data

The data for this study were selected from the 2020 and 2022 waves of the China Family Panel Studies (CFPS), a national, comprehensive, and large-scale research survey initiated by the Institute of Social Science Survey (ISSS), Peking University. By tracking data at the individual, household, and community levels, CFPS reflects changes in China's society, economy, population, and health. The survey employs stratified sampling, with respondent samples covering 25 provinces/autonomous regions in China, ensuring relatively strong representativeness of the data. Since its launch in 2008, CFPS has completed 7 rounds of surveys as of 2024, with the most recent round conducted in 2022. Given the rapid development of the internet in China, to ensure the timeliness and reliability of research conclusions, this study uses data from the two most recent CFPS waves. Compared with earlier years, the 2020 and 2022 surveys include questions on the extent to which rural residents experience the digital utility divide, enabling a deeper exploration of the impact of the

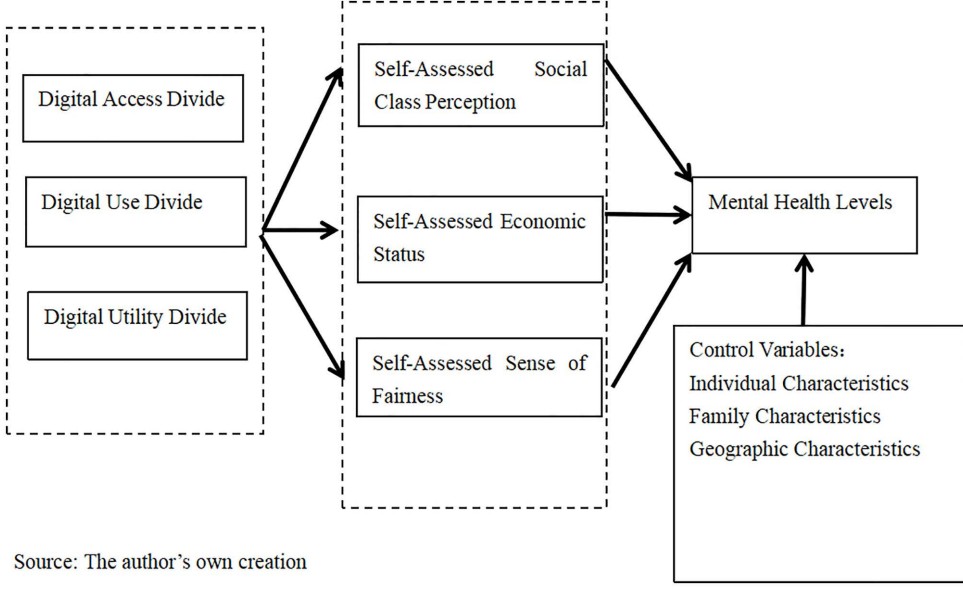

Source: The author's own creation

**Fig 1. The research logic of this article.**

digital divide on the mental health of China's rural residents, which is of special and profound significance for the investigation.The China Family Panel Studies (CFPS) has been approved by the Institutional Review Board of Peking University (approval No. IRB00001052-14010). All respondents provided written informed consent prior to data collection. The CFPS data are de-identified and publicly released by the Institute of Social Science Survey at Peking University. As this study uses only anonymized secondary data, no additional ethical review is required.

The initial target sample size for CFPS2022 was 16,000 households, using implicit stratified sampling and multi-stage equal probability sampling. Specifically, administrative regions were first selected as primary sampling units. Second, administrative villages were selected as sampling units based on the first step. Finally, households were selected as the terminal sampling units. During sampling, researchers constructed the sampling frame using the map address method and adopted cyclic sampling with random starting points to ensure the reliability of sampling results.

In our study, we carefully handled missing data and considered the use of sample weights to ensure the robustness and representativeness of our analyses. Missing data were addressed through stringent procedures to preserve data integrity. Specifically, we excluded observations that were not successfully matched across the 2020 and 2022 waves of the China Family Panel Studies (CFPS) and dropped cases with missing values on key independent or dependent variables, yielding a final analytic sample of 11,114 valid observations.

Regarding weights, the CFPS data set does not provide sampling weights; consequently, our analyses rely on unweighted data. We have evaluated the representativeness of our sample and consider it to adequately reflect the characteristics of rural Chinese residents, given that CFPS employs a stratified sampling design covering 25 provinces/autonomous regions. To mitigate potential biases arising from the absence of sampling weights and from item non-response, we control for a comprehensive set of individual- and household-level covariates and conduct extensive robustness checks. These measures jointly enhance the reliability and validity of our findings. Moreover, a number of published studies likewise omit weight adjustments and still obtain widely accepted results; in causal-inference settings such as ours, the influence of weighting is typically of secondary importance.

## 3.2. Psychometric properties of composite variables

Principal-component analysis (PCA) was used to construct composite indices for mental health, digital use, and digital utility (Table 1). The mental-health scale was derived from eight items; four principal components were retained, explaining 74.3% of the total variance (first component = 39.9%). Item loadings ranged from 0.24 to 0.43. Cronbach's α = 0.80, indicating good internal consistency; mean inter-item correlation = 0.30 (range 0.17–0.52). The composite score was approximately normal (M = 0.00, SD = 1.00). The digital-use-divide index was derived from four items; three principal components were retained, explaining 86.2% of the variance (first component = 36.5%). Loadings ranged from 0.15 to 0.69. Cronbach's α = 0.35, denoting modest reliability; mean inter-item correlation = 0.06 (range 0.04–0.11). The composite was roughly normal (M = 0.00, SD = 0.66). The digital-utility-divide index was derived from five items; the retained components explained 79.2% of the variance (first component = 36.5%). Loadings ranged from 0.26 to 0.52. Cronbach's α = 0.73, indicating good internal consistency; mean inter-item correlation = 0.36 (range 0.16–0.50). The composite score was approximately normal (M = 0.00, SD = 0.98). Table 2 presents the detailed component loadings for each item. For mental health, "felt down" and "everything an effort" loaded positively on the first component, contributing to the overall score. For digital use, "online gaming" and "online learning" showed salient loadings on the first component. For digital utility, "importance for work" and "importance for entertainment" loaded on the first component. These patterns support the factorial validity of the indices, indicating that the items adequately capture the intended constructs.

**Table 1. Psychometric properties of PCA-based composites.**

| Construct | Items | Retained PCs | Cumulative variance (%) | Cronbach α | Loading range | Item correlations | Composite M ± SD |
|---|---|---|---|---|---|---|---|
| Mental Health Levels | 8 | 4 | 74.3 | 0.78 | 0.24-0.43 | 0.30 (0.17–0.52) | 0.00 ± 1.00 |
| Digital Use Divide | 4 | 3 | 86.2 | 0.35 | 0.15–0.69 | 0.06 (0.04–0.11) | 0.00 ± 0.66 |
| Digital Utility Divide | 5 | 3 | 79.2 | 0.73 | 0.26–0.52 | 0.36 (0.16–0.50) | 0.00 ± 0.98 |

**Table 2. Loadings for PCA-based composites.**

| Item | Construct | Comp1 | Comp2 | Comp3 | Comp4 | Unexplained |
|---|---|---|---|---|---|---|
| How often have you felt pleased in the past week | Mental health | 0.237 | 0.668 | 0.122 | 0.012 | 0.219 |
| How often have you felt happy in life in the past week | Mental health | 0.258 | 0.646 | 0.027 | 0.008 | 0.235 |
| How often have you felt down in the past week? | Mental health | 0.381 | −0.154 | 0.323 | −0.553 | 0.211 |
| How often have you felt that everything is an effort in the past week | Mental health | 0.381 | −0.209 | 0.400 | −0.305 | 0.289 |
| How often have you had trouble falling asleep in the past week | Mental health | 0.329 | −0.185 | 0.651 | 0.625 | 0.020 |
| How often have you felt lonely in the past week | Mental health | 0.390 | −0.125 | −0.311 | 0.195 | 0.411 |
| How often have you felt sad in the past week | Mental health | 0.426 | −0.089 | −0.355 | 0.013 | 0.311 |
| How often have you felt sad in the past week | Mental health | 0.382 | −0.103 | −0.516 | 0.068 | 0.306 |
| Do you play online games daily | Digital use | 0.149 | 0.694 | −0.701 | – | 0.003 |
| Do you use the internet for learning daily | Digital use | 0.695 | −0.122 | −0.048 | – | 0.277 |
| Do you use short videos daily | Digital use | 0.688 | −0.175 | 0.047 | – | 0.273 |
| Do you shop online daily | Digital use | 0.148 | 0.688 | 0.710 | – | 0.000 |
| How important is internet learning for work | Digital utility | 0.485 | −0.236 | 0.055 | – | 0.211 |
| How important is the internet for entertainment | Digital utility | 0.452 | −0.128 | 0.782 | – | 0.096 |
| How important is the internet for keeping in touch with family and friends | Digital utility | 0.264 | 0.956 | 0.055 | – | 0.012 |
| How important is internet learning for work | Digital utility | 0.515 | −0.096 | −0.241 | – | 0.315 |
| How important is the internet for daily life | Digital utility | 0.476 | −0.066 | −0.570 | – | 0.239 |

### 3.3. Variables

**3.3.1. Dependent variable: Level of mental health.** As previously noted, this study follows Li's multidimensional approach to assessing respondents' mental-health status, drawing on the relevant battery of items contained in the CFPS2020 and CFPS2022 questionnaires [53]. The items ask how often during the past week respondents experienced specific feelings such as "felt down," "felt lonely," "felt sad," as well as positive affects "felt pleased" and "felt happy." Answers are recorded on a four-point scale ranging from "most of the time (5–7 days)" to "almost never (<1 day)," coded 1–4. A composite mental-health score was created by principal-component analysis (PCA), with higher values denoting better mental health. To ensure consistent polarity, the two positive-affect items ("felt pleased" and "felt happy") were reverse-coded so that higher scores reflect more frequent positive affect.

**3.3.2. Independent variable: Digital divide.** The CFPS 2020 and CFPS 2022 datasets include information on respondents' internet access, usage, and utility. Based on these data, we followed the methods for measuring different levels of the digital divide as outlined in existing research and employed the framework of "digital access divide—digital use divide—digital utility divide" to measure the digital divide among rural residents in China [54,55]. The specific details of this framework are presented below.

Digital Access Divide.The first-level digital divide is measured by assessing individuals' access to the internet. In this study, respondents were asked two questions: "Do you use mobile phones or other mobile devices to access the internet?" and "Do you use computers to access the internet?" If a respondent answered "No" to both questions, they were considered to be affected by the digital access divide. Based on this, a binary variable "digital access divide" was created, where 1 indicates the presence of the digital access divide and 0 indicates its absence.

For the Digital Use Divide,the indicator is constructed from respondents' frequency and purpose of internet use. Among those not affected by the digital access divide, the questionnaire records whether they engage in four online activities (online gaming, online learning, short-video viewing, and online shopping). A "yes" response is coded 0 (no use divide), whereas "no" is coded 1 (divide present). These four items were submitted to principal-component analysis (PCA) to create the "digital use divide" variable; higher values signify a more pronounced digital use divide.

The digital utility divide is determined by the perceived importance of the internet. For respondents unaffected by the digital access divide, the questionnaire assesses the importance of the internet in five domains: learning, work, staying in touch with family and friends, entertainment, and daily life. Each item is rated on a five-point scale from "very important" to "very unimportant," coded 1–5. These five items were combined via principal-component analysis (PCA) into a single "digital utility divide" variable; higher values indicate lower perceived importance of the internet and thus a more severe digital utility divide.

**3.3.3. Control variables.** Referring to existing literatur and in line with the purpose and content of this study, the following control variables were selected [9,56].

Individual Characteristics:

Gender:Respondents were asked, "What is your gender (male or female)?" A binary dummy variable "gender" was created, with 1 for males and 0 for females.

Education:Respondents were asked, "What is the highest level of education you have completed?" A binary dummy variable "edu" was created, with 1 for those with an associate degree or higher, and 0 otherwise.

Marital Status:Respondents were asked, "What is your current marital status (unmarried, married, widowed, divorced)?" A binary dummy variable "marriage" was created, with 1 for those who are married, and 0 otherwise.

Age:Respondents were asked, "How old are you?" A continuous variable "age" was created based on the responses, with values assigned according to the reported age.

Religion:Respondents were asked, "Are you a member of a religious group?" A binary dummy variable "religion" was created, with 1 for those who answered yes, and 0 otherwise.

Family Characteristics:

Family Size:Respondents were asked, "How many members are there in your household?" A variable "fml_count" was created to describe the size of the family.

Housing:Respondents were asked, "What is the value of your current housing (in yuan)?" A binary dummy variable "house" was created, with 0 for those with housing value of 0 (no housing), and 1 for those who own housing.

Car Ownership:Respondents were asked, "Does your household own a car (yes/no)?" A binary dummy variable "car" was created, with 1 for those who answered yes, and 0 otherwise.

Geographic Characteristics:

To account for geographic characteristics and economic development differences, dummy variables "region1" and "region2" were created. "Region1" uses the central region as a reference and assigns a value of 1 if the respondent's province is in the eastern region, and 0 otherwise. "Region2" also uses the central region as a reference and assigns a value of 1 if the respondent's province is in the western region, and 0 otherwise, to explore the impact of geographic characteristics.

**3.3.4. Mediating variables.** Drawing on related literature and focusing on self-assessed social class perception, self-assessed economic status, and self-assessed sense of fairness, this study investigates the mechanisms through which the digital divide affects the mental health of rural residents in China [53,57].

Self-Assessed Social Class Perception:Respondents were asked, "On a scale of 1 to 5, how would you rate your social status in your local area?" A variable "class" was created to describe respondents' self-assessed social class perception, with higher values indicating a higher perceived social class.

Self-Assessed Economic Status:Respondents were asked, "On a scale of 1 to 5, how would you rate your income level in your local area?" A variable "condition" was created to describe self-assessed economic status, with higher values indicating a higher perceived economic status.

Self-Assessed Sense of Fairness:Respondents were asked, "On a scale of 0 to 10, how would you rate the wealth gap in our country?" A variable "fair" was created to measure respondents' self-assessed sense of fairness, with higher values indicating a higher perceived level of social fairness.

Descriptive analyses are presented in Table 3.

# 4. Empirical analysis

## 4.1. Benchmark results

Because several independent variables as well as the dependent variable are constructed with principal-component analysis and the data are the 2020–2022 CFPS panel, we estimate a two-way fixed-effects model that simultaneously controls for individual-specific and year-specific unobserved heterogeneity, thereby mitigating omitted-variable bias. The specification is:

$$\text{Mental Health Levels}_{it} = C + \alpha_i + \lambda_t + \beta \text{ digital divide}_{it} + \gamma X_{it} + \varepsilon_{it} \tag{1}$$

In Equation (1), "Mental Health Levels" is the dependent variable, representing mental health status. It is derived from principal component analysis (PCA) of collected questions, resulting in a variable ranging from −7.7752 to 2.3401. This variable represents the mental health of individual i at time t, with higher values indicating better mental health. C is the constant term.

"digital divide$_{it}$" is the independent variable, representing the digital divide score for individual i at time t, which includes the digital access divide, digital use divide, and digital utility divide. $\alpha_i$ is the individual-specific intercept that controls for the impact of individual characteristics on mental health. $\lambda_t$ is the time-specific intercept that accounts for time-related influences on mental health. $\beta$ represents the regression coefficient, and $\gamma$ denotes the coefficients of the control variables. $X_{it}$ refers to a set of control variables, and $\varepsilon_{it}$ is the error term.

**Table 3. Descriptive analysis.**

| Item | Opt | Cnt | N | Pet |
|------|-----|-----|---|-----|
| Mental Health Levels | | | | |
| How often have you felt pleased in the past week | less than 1 day | 868 | 11114 | 7.81 |
| | 1–2 days | 2900 | 11114 | 26.09 |
| | 3–4 days | 3435 | 11114 | 30.91 |
| | 5–7 days | 3911 | 11114 | 35.19 |
| How often have you felt happy in life in the past week | less than 1 day | 717 | 11114 | 6.45 |
| | 1–2 days | 2414 | 11114 | 21.81 |
| | 3–4 days | 3630 | 11114 | 32.66 |
| | 5–7 days | 4353 | 11114 | 39.17 |
| How often have you felt down in the past week? | less than 1 day | 4378 | 11114 | 39.39 |
| | 1–2 days | 5092 | 11114 | 45.82 |
| | 3–4 days | 1021 | 11114 | 9.19 |
| | 5–7 days | 623 | 11114 | 5.61 |
| How often have you felt that everything is an effort in the past week | less than 1 day | 4704 | 11114 | 42.32 |
| | 1–2 days | 4364 | 11114 | 39.27 |
| | 3–4 days | 1180 | 11114 | 10.62 |
| | 5–7 days | 866 | 11114 | 7.79 |
| How often have you had trouble falling asleep in the past week | less than 1 day | 5040 | 11114 | 45.35 |
| | 1–2 days | 3615 | 11114 | 32.53 |
| | 3–4 days | 1491 | 11114 | 13.42 |
| | 5–7 days | 968 | 11114 | 8.71 |
| How often have you felt lonely in the past week | less than 1 day | 6544 | 11114 | 58.88 |
| | 1–2 days | 3303 | 11114 | 29.72 |
| | 3–4 days | 708 | 11114 | 6.37 |
| | 5–7 days | 559 | 11114 | 5.03 |
| How often have you felt sad in the past week | less than 1 day | 5854 | 11114 | 52.67 |
| | 1–2 days | 4241 | 11114 | 38.16 |
| | 3–4 days | 627 | 11114 | 5.64 |
| | 5–7 days | 392 | 11114 | 3.53 |
| How often have you felt sad in the past week | less than 1 day | 8681 | 11114 | 78.11 |
| | 1–2 days | 1797 | 11114 | 16.17 |
| | 3–4 days | 322 | 11114 | 2.90 |
| | 5–7 days | 314 | 11114 | 2.83 |
| Digital Divide | | | | |
| Digital Access Divide | NO = 1 | 6826 | 11114 | 61.42 |
| | YES = 0 | 4288 | 11114 | 38.58 |
| Digital Use Divide | | | | |
| Do you play online games daily | NO = 1 | 5248 | 6815 | 77.01 |
| | YES = 0 | 1567 | 6815 | 22.99 |
| Do you use the internet for learning daily | NO = 1 | 4349 | 6815 | 63.82 |
| | YES = 0 | 2466 | 6815 | 36.18 |
| Do you use short videos daily | NO = 1 | 1876 | 6815 | 27.53 |
| | YES = 0 | 4939 | 6815 | 72.47 |
| Do you shop online daily | NO = 1 | 3160 | 6815 | 46.37 |
| | YES = 0 | 3655 | 6815 | 53.63 |

*(Continued)*

**Table 3.** (Continued)

| Item | Opt | Cnt | N | Pet |
|---|---|---|---|---|
| Digital Utility Divide | | | | |
| How important is internet learning for work | 1 | 2350 | 6815 | 34.48 |
| | 2 | 1273 | 6815 | 18.68 |
| | 3 | 1873 | 6815 | 27.48 |
| | 4 | 588 | 6815 | 8.63 |
| | 5 | 731 | 6815 | 10.73 |
| How important is the internet for entertainment | 1 | 1703 | 6815 | 24.99 |
| | 2 | 1532 | 6815 | 22.48 |
| | 3 | 2436 | 6815 | 35.74 |
| | 4 | 737 | 6815 | 10.81 |
| | 5 | 407 | 6815 | 5.97 |
| How important is the internet for keeping in touch with family and friends | 1 | 3858 | 6815 | 56.61 |
| | 2 | 2458 | 6815 | 36.07 |
| | 3 | 379 | 6815 | 5.56 |
| | 4 | 74 | 6815 | 1.09 |
| | 5 | 46 | 6815 | 0.67 |
| How important is internet learning for work | 1 | 2565 | 6815 | 37.64 |
| | 2 | 1742 | 6815 | 25.56 |
| | 3 | 1599 | 6815 | 23.46 |
| | 4 | 478 | 6815 | 7.01 |
| | 5 | 431 | 6815 | 6.32 |
| How important is the internet for daily life | 1 | 2637 | 6815 | 38.69 |
| | 2 | 1467 | 6815 | 21.53 |
| | 3 | 1533 | 6815 | 22.49 |
| | 4 | 611 | 6815 | 8.97 |
| | 5 | 567 | 6815 | 8.32 |
| Mediating Variables | | | | |
| Class | | | | |
| How would you rate your social status in your local area | 1 | 865 | 11114 | 7.78 |
| | 2 | 1583 | 11114 | 14.24 |
| | 3 | 5030 | 11114 | 45.26 |
| | 4 | 2069 | 11114 | 18.62 |
| | 5 | 1567 | 11114 | 14.10 |
| Condition | | | | |
| How would you rate your income level in your local area | 1 | 1698 | 11114 | 15.28 |
| | 2 | 1836 | 11114 | 16.52 |
| | 3 | 4942 | 11114 | 44.47 |
| | 4 | 1433 | 11114 | 12.90 |
| | 5 | 1205 | 11114 | 10.84 |

*(Continued)*

**Table 3.** (Continued)

| Item | Opt | Cnt | N | Pet |
|---|---|---|---|---|
| Fair | | | | |
| How would you rate the wealth gap in our country | 0 | 224 | 11114 | 2.02 |
| | 1 | 74 | 11114 | 0.67 |
| | 2 | 131 | 11114 | 1.18 |
| | 3 | 366 | 11114 | 3.29 |
| | 4 | 277 | 11114 | 2.49 |
| | 5 | 1729 | 11114 | 15.56 |
| | 6 | 1381 | 11114 | 12.43 |
| | 7 | 1024 | 11114 | 9.22 |
| | 8 | 1214 | 11114 | 10.92 |
| | 9 | 338 | 11114 | 3.04 |
| | 10 | 1466 | 11114 | 13.19 |
| Control Variables | | | | |
| Gender | Female = 0 | 5442 | 11114 | 48.97 |
| | Male = 1 | 5672 | 11114 | 51.03 |
| Edu | Above senior high school = 0 | 708 | 11114 | 6.37 |
| | Below senior high school = 1 | 10406 | 11114 | 93.63 |
| Marriage | NO = 0 | 1609 | 11114 | 14.48 |
| | YES = 1 | 9505 | 11114 | 85.52 |
| Religion | NO = 0 | 10824 | 11114 | 97.39 |
| | YES = 1 | 290 | 11114 | 2.61 |
| House | NO = 0 | 3953 | 11114 | 35.57 |
| | YES = 1 | 7161 | 11114 | 64.43 |
| Car | NO = 0 | 8011 | 11114 | 72.08 |
| | YES = 1 | 3103 | 11114 | 27.92 |
| Region1 | Non-eastern region = 0 | 6758 | 11114 | 60.81 |
| | Eastern region = 1 | 4356 | 11114 | 39.19 |
| Region2 | Non-western region = 0 | 7304 | 11114 | 65.72 |
| | Western region = 1 | 3810 | 11114 | 34.28 |

| Item | N | Mean, | Median | Minimum | Maximum | Standard Deviation | Kurtosis | Skewness |
|---|---|---|---|---|---|---|---|---|
| age | 11114 | 46.8597 | 48 | 16 | 89 | 16.3172 | 2.0435 | −0.0569 |
| fml_count | 11114 | 4.3644 | 4 | 1 | 15 | 2.0921 | 4.0849 | 0.7619 |
| Digital Use Divide | 6815 | −4.43e−17 | 0.0808 | −1.0962 | 1.1415 | 0.6576 | 2.0881 | 0.1105 |
| Digital Utility Divide | 6815 | −2.95e−17 | −0.0722 | −1.4784 | 4.1687 | 0.9797 | 3.5166 | 0.6349 |
| Mental Health Levels | 11114 | −1.09e−16 | 0.0967 | −4.1124 | 1.4671 | 0.9954 | 3.2657 | −0.6532 |

Table 4 shows the impact of the digital divide at different levels on the mental health of rural Chinese residents, with results presented in columns (1) to (3). As indicated in Table 2, the digital access divide has a negative effect on mental health at the 1% significance level, suggesting that rural residents without internet access have lower mental health levels. The digital use divide also shows a negative impact at the 1% level, indicating that less frequent internet use correlates with lower mental health. Similarly, the digital utility divide is significantly negative at the 1% level, implying that when rural residents perceive the internet as less important, their mental health is negatively affected. Thus, Hypothesis H1 is supported.

**Table 4. Benchmark regression.**

| Item | (1) | (2) | (3) |
|---|---|---|---|
| Digital Access Divide | −0.1100*** (−2.63) | | |
| Digital Use Divide | | −0.1430*** (−6.25) | |
| Digital Utility Divide | | | −0.1090*** (−6.40) |
| Gender | −0.1370 (−0.27) | 0.4740 (0.67) | 0.4770 (0.67) |
| Edu | −0.2240* (−1.76) | −0.1800 (−1.38) | −0.1830 (−1.41) |
| Marriage | 0.2290* (1.89) | 0.3740*** (2.86) | 0.3620*** (2.77) |
| Age | 0.0010 (0.15) | −0.0020 (−0.30) | −0.004 (−0.52) |
| Religion | −0.0135 (−0.18) | 0.0984 (0.83) | 0.0766 (0.65) |
| Fml_count | 0.0193* (1.96) | 0.0158 (1.29) | 0.0160 (1.31) |
| House | 0.1500** (2.17) | 0.0261 (0.26) | 0.0318 (0.32) |
| Car | 0.1030* (1.65) | 0.1580* (1.91) | 0.166** (2.02) |
| Region1 | Control | | |
| Region2 | Control | | |
| N | 11114 | 6815 | 6815 |

Notes:① *, **, and *** denote significance at the 10%, 5%, and 1% levels, respectively.

② t-statistics are in parentheses.

③ Individual and time fixed effects are controlled for.

As shown in columns (1) to (3) of Table 4, among the control variables, variables like car ownership are consistently insignificant, while education and marriage have significant effects. Variables such as family size and housing are significant only in certain digital divide contexts.To sharpen the estimated impact of each digital-divide layer on rural residents' mental health, Table 5 reports 95% confidence intervals and standardized coefficients. Columns (1)–(3) show that the 95% CIs for the digital access divide (−0.1959, −0.0237), the digital use divide (−0.1886, −0.0982) and the digital utility divide (−0.1468, −0.0703) are entirely below zero, underscoring the precision and reliability of the negative effects. Columns (4)–(6) present standardized coefficients that permit effect-size comparisons across metrics. The standardized beta for the digital access divide is −0.0537 ($p < 0.05$), indicating a medium effect. The corresponding values for the digital use divide and the digital utility divide are −0.0946 and −0.1070 (both $p < 0.01$), respectively. Thus, the digital utility divide exerts the largest relative influence, followed by the digital use divide, while the digital access divide shows the smallest standardized impact. These results imply that—although all three layers harm mental health—interventions that heighten rural residents' perceived value of internet use (i.e., reduce the digital utility divide) are likely to yield the largest mental-health gains. Standardized coefficients therefore offer a clear empirical basis for prioritizing policies that enhance perceived internet utility alongside continued efforts to improve access and usage skills.

**Table 5. Digital divide's impact on rural residents' mental health (with 95% CIs & Std. Coeff.).**

| Item | (1) | (2) | (3) | (4) | (5) | (6) |
|---|---|---|---|---|---|---|
| Digital Access Divide | [−0.1959,−0.0237] | | | −0.0537** (−2.50) | | |
| Digital Use Divide | | [−0.1886, −0.0982] | | | −0.0946*** (−6.22) | |
| Digital Utility Divide | | | [−0.1468, −0.0703] | | | −0.1070*** (−5.57) |
| Gender | [−0.8866,0.6129] | [−0.3454, 1.2943] | [−0.1825, 1.1367] | −0.1380 (−0.36) | 0.4760 (1.13) | 0.4790 (1.42) |
| Edu | [−0.4456,−0.0032] | [−0.4240, 0.0639] | [−0.4228, 0.0560] | −0.2250** (−1.99) | −0.1810 (−1.45) | −0.1840 (−1.50) |
| Marriage | [−0.0261, 0.4833] | [0.0987, 0.6489] | [0.0893, 0.6352] | 0.2300* (1.76) | 0.3750*** (2.66) | 0.3630*** (2.60) |
| Age | [−0.0117, 0.0137] | [0.0987, 0.6489] | [−0.0216, 0.0126] | 0.0010 (0.15) | −0.0027 (−0.30) | −0.004 (−0.52) |
| Religion | [−0.1641, 0.1371] | [−0.1755, 0.3723] | [−0.1978, 0.3510] | −0.0136 (−0.18) | 0.0987 (0.70) | 0.0768 (0.55) |
| Fml_count | [−0.0007, 0.0393] | [−0.0093, 0.0409] | [−0.0090, 0.0410] | 0.0193* (1.89) | 0.0159 (1.24) | 0.0161 (1.25) |
| House | [0.0067, 0.2932] | [−0.1782, 0.2305] | [−0.1736, 0.2373] | 0.1510** (2.05) | 0.0262 (0.25) | 0.0319 (0.30) |
| Car | [−0.0258, 0.2326] | [−0.0234, 0.3387] | [−0.0133, 0.3463] | 0.1040 (1.57) | 0.158* (1.71) | 0.167* (1.82) |
| Region1 | Control | | | | | |
| Region2 | Control | | | | | |
| N | 11114 | 6815 | 6815 | 11114 | 6815 | 6815 |

Notes:

① *, **, and *** denote significance at the 10%, 5%, and 1% levels, respectively.

② Columns (1)–(3) report 95% confidence intervals of the estimated coefficients to assess precision.

③ Columns (4)–(6) report standardized coefficients to facilitate comparison of effect sizes across variables.

④ Individual and year fixed effects are controlled for.

## 4.2. Robustness tests

Robustness Test 1: Replacing the Dependent Variable. Instead of using principal component analysis (PCA) to measure mental health levels based on responses to questions like "How often have you felt down in the past week?", the mental health level was measured using the average of responses to these questions, with results shown in Table 6.

Robustness Test 2: Replacing the Independent Variable. The digital access divide was measured by the question "Do you use mobile devices to access the internet?". The digital use and utility divides were measured using averages instead of PCA, with results presented in Table 6.

Robustness Test 3: Replacing Control Variables. Education was redefined on a scale from 1 (below secondary vocational school) to 6 (doctorate). The central and western regions were combined into a non-eastern region (coded as 0), with the eastern region coded as 1. Family size was redefined as 1 for families with ≤6 members and 2 for families with >6 members. Results are in Table 6.

Robustness Test 4: Replacing the Econometric Model. The two-way fixed-effects model was replaced with a random-effects model, and the results are shown in Table 6.

Robustness Test 5: Trimming Outliers. Samples in the top and bottom 5% of the age distribution were trimmed, with results presented in Table 6.

**Table 6. Robustness tests.**

| Robustness Test 1: Replacing the Dependent Variable. | (1) | (2) | (3) |
|---|---|---|---|
| Digital Access Divide | −0.0569** (−2.57) | | |
| Digital Use Divide | | −0.0498*** (−9.35) | |
| Digital Utility Divide | | | −0.0181*** (−3.33) |
| N | 11114 | 6815 | 6815 |
| Robustness Test 2: Replacing the Independent Variable. | | | |
| Digital Access Divide | 0.2060*** (2.75) | | |
| Digital Use Divide | | −0.5200*** (−5.77) | |
| Digital Utility Divide | | | −0.1000*** (−2.84) |
| N | 11114 | 6815 | 6815 |
| Robustness Test 3: Replacing Control Variables. | | | |
| Digital Access Divide | −0.111*** (−2.63) | | |
| Digital Use Divide | | −0.143*** (−6.22) | |
| Digital Utility Divide | | | −0.109*** (−6.41) |
| N | 11114 | 6815 | 6815 |
| Robustness Test 4: Replacing the Econometric Model. | | | |
| Digital Access Divide | −0.124*** (−4.73) | | |
| Digital Use Divide | | −0.171*** (−9.67) | |
| Digital Utility Divide | | | −0.108*** (−9.02) |
| N | 11114 | 6815 | 6815 |
| Robustness Test 5: Trimming Outliers. | | | |
| Digital Access Divide | −0.111*** (−2.66) | | |
| Digital Use Divide | | −0.143*** (−6.25) | |
| Digital Utility Divide | | | −0.109*** (−6.40) |
| N | 11114 | 6815 | 6815 |

Notes: ① *, **, and *** denote significance at the 10%, 5%, and 1% levels, respectively.

② t-statistics are in parentheses.

③ Individual and time fixed effects are controlled for.

④ Control variables have been accounted for.

In summary, as shown in Table 3, the digital divide at all levels (access, use, and utility) remains significantly negatively associated with the mental health of rural Chinese residents, regardless of whether the dependent variable, independent variable, control variables, or econometric model are replaced, or whether outliers are trimmed. This confirms the robustness of the benchmark regression results in Table 6.

## 4.3. Endogeneity test

When exploring the impact of the digital divide on the mental health of rural Chinese residents, researchers must carefully address endogeneity issues to prevent systematic biases in research results.

First, the problem of omitted variables in model specification requires attention. Potential factors such as individual psychological qualities may simultaneously influence both the digital divide and mental health. However, due to researchers' oversight, these factors are often ignored, ultimately leading to deviations between research results and reality. This study mitigates this issue by incorporating as many control variables as possible to reduce biases caused by omitted variables.

Second, measurement errors in data can also affect the robustness of final results. Respondents' self-reports on internet usage and its consequences may contain biases stemming from cognitive differences, memory errors, or social

desirability. Similarly, respondents' evaluations of their own mental health levels may be inaccurate, prone to emotional fluctuations or social comparison effects. Effectively identifying and correcting measurement errors is a key problem that must be solved in empirical research.

Finally, the presence of bidirectional causality may exacerbate endogeneity. Bidirectional causality means that the digital divide not only affects the mental health of rural residents but also that their mental health status may in turn influence their evaluations of internet access, usage, and utility, impacting the accuracy of research results. To address this, this study employs the instrumental variable (IV) method. Drawing on research by Li and Peng, we use the mean value of the digital divide among the same age group in the respondent's village as the instrumental variable [54,58]. For example, when testing the endogeneity of the digital access divide, we use the mean value of the digital access divide among the same age group in the respondent's village as the IV. The theoretical justification for treating the village-level, same-age-cohort mean of each digital-divide dimension as an exogenous shock rests on the supply-side determinants of rural internet access. Experimental evidence shows that China's "Universal Service" fibre-to-the-village programme allocated roll-out quotas to administrative villages by administrative decree, generating discontinuous jumps in network availability that are orthogonal to household-level unobservables [59]. Because the "last-mile" investment is negotiated by village committees and subsidised by county governments, the proportion of a specific age group within a village that remains unconnected, uses the internet infrequently, or attributes low utility to it is driven primarily by these exogenous infrastructure thresholds rather than by individual psycho-social traits. The cohort mean therefore captures a community-level supply shock that satisfies the relevance condition and is systematically unrelated to the idiosyncratic component of mental-health outcomes.

Second, social-norms and reference-group theory provide complementary support for the exclusion restriction. Zhang demonstrate that, once network coverage is in place, peer visibility within the village creates a threshold effect: adoption accelerates as soon as the peer-use rate crosses a critical point, independently of individual socioeconomic status [60]. Since beliefs about mental health in rural China are formed through local rather than cross-village comparisons, the village–age-group mean influences psychological well-being only by shaping an individual's own digital access, intensity of use, and perceived utility. The three instruments thus isolate exogenous community-level variation in the digital access, use, and utility divides, enabling causal identification of each divide's effect on mental health.

The IV results are reported in Table 7. After instrumenting, the digital access divide continues to exert a negative effect on rural residents' mental health. In the first stage the F-statistic equals 723.72—far above the conventional threshold

**Table 7. Endogeneity test.**

| Item | (1) | (2) | (3) |
|---|---|---|---|
| Digital Access Divide | −0.0880* (−1.57) | | |
| Digital Use Divide | | −0.158*** (−5.74) | |
| Digital Utility Divide | | | −0.105*** (−5.10) |
| First-stage regression coefficient | 0.824*** (83.99) | 0.976*** (84.50) | 0.910*** (80.32) |
| Hausman test value | 0.35 | 0.92 | .0.08 |
| Partial R2 | 0.7013 | 0.7425 | 0.7534 |
| First-stage F-statistic | 723.72 | 727.43 | 657.38 |
| p-value | 1.00 | 0.99 | 1.00 |
| N | 11114 | 6815 | 6815 |

Notes: ① *, **, and *** denote significance at the 10%, 5%, and 1% levels, respectively.

② t-statistics are in parentheses.

③ Individual and time fixed effects are controlled for.

④ Control variables have been accounted for.

of 10—so the weak-instrument test is easily passed. The instrument is positively and significantly related to the access divide at the 1% level, confirming relevance. The Hausman statistic is 0.02 with a p-value of 1.00, indicating that the access divide can be treated as exogenous. The IV estimate remains negative and significant at the 10% level, corroborating robustness; the first-stage partial $R^2$ is 0.7013, showing that the instrument explains a large share of the variation in the endogenous regressor.

For the digital use divide, the first-stage coefficient is 0.9890 (significant at 1%), and the F-statistic is 718.68, again rejecting the weak-instrument hypothesis. The second-stage coefficient is negative and significant at 1%. The Hausman test yields 1.40 (p = 0.9992), so the use divide is also exogenous. The partial $R^2$ reaches 0.7425, reaffirming instrument strength.

Turning to the digital utility divide, the first-stage coefficient equals 0.8980 (significant at 1%), with an F-statistic of 615.12. The IV estimate is −0.0427 and significant at 10%. The Hausman statistic is 0.29 (p = 1.00), and the partial $R^2$ is 0.7534, demonstrating strong explanatory power.

Taken together, the instruments perform well: partial $R^2$ values of 0.7013, 0.7425 and 0.7534 imply a high correlation with the respective endogenous variables, while the large first-stage F-statistics and high Hausman p-values jointly establish instrument exogeneity and validity. Thus, the baseline findings are robust to endogeneity concerns: all three layers of the digital divide significantly undermine the mental health of rural Chinese residents.

## 4.4. Sub-dimension analysis

The disaggregated results in Table 8 show that the frequency with which rural residents play online games has no discernible effect on their mental-health score. By contrast, skills related to online learning, e-shopping, and the use of new-media platforms (short videos) are all positively and significantly associated with better mental health; respondents who report lower proficiency in these domains exhibit markedly poorer psychological well-being. Although online gaming has expanded rapidly and is widely recognised, rural residents—who on average have lower educational attainment and limited digital literacy—often lack the competencies needed to engage meaningfully with games; this plausibly explains the null finding for gaming frequency. The remaining dimensions of the digital-use divide are therefore more salient for rural residents, because they govern the ability to access information safely and conveniently and to improve daily living. Deficiencies in these practical skills translate directly into lower mental-health scores.

Table 9 turns to the sub-dimensions of the digital-utility divide. Perceived importance of the internet for "staying in touch with family and friends" is negatively associated with mental health at the 1% level: the higher the subjective value placed on online interpersonal contact, the worse the psychological outcome. Utilities attached to work, entertainment, learning

**Table 8. Sub-dimension analysis of the digital use divide.**

| Item | (1) | (2) | (3) | (4) |
|---|---|---|---|---|
| Digital Use Divide | | | | |
| Frequency of online gaming | −0.0414 (−0.96) | | | |
| Online learning skills | | −0.417*** (−15.28) | | |
| Online shopping skills | | | 0.0563* (1.75) | |
| Short video skills | | | | −0.0806*** (−2.71) |
| N | 6815 | 6815 | 6815 | 6815 |

Notes: ① *, **, and *** denote significance at the 10%, 5%, and 1% levels, respectively.

② t-statistics are in parentheses.

③ Individual and time fixed effects are controlled for.

④ Control variables have been accounted for.

**Table 9. Sub-dimension analysis of the digital utility divide.**

| Item | (1) | (2) | (3) | (4) | (5) |
|---|---|---|---|---|---|
| Digital Utility Divide | | | | | |
| Work | −0.0115 (−0.96) | | | | |
| Leisure and entertainment | | −0.0169 (−1.23) | | | |
| Maintaining contact with family and friends | | | −0.222*** (−11.51) | | |
| Learning | | | | −0.00571 (−0.43) | |
| Daily life | | | | | −0.0122 (−0.96) |
| N | 6815 | 6815 | 6815 | 6815 | 6815 |

Notes: ① *, **, and *** denote significance at the 10%, 5%, and 1% levels, respectively.

② t-statistics are in parentheses.

③ Individual and time fixed effects are controlled for.

④ Control variables have been accounted for.

and daily life, however, are statistically insignificant. A likely explanation is the socio-economic context of rural China: villages are relatively isolated and economically under-developed, prompting many working-age adults to migrate to cities. When the internet fails to deliver the expected relational benefits—i.e., when the utility divide in the social-connectivity domain is wide—feelings of loneliness intensify and mental health deteriorates. Lower average schooling and lagging local economic development may also attenuate the psychological relevance of internet utilities outside the interpersonal sphere, producing the non-significant results observed for the other domains.

## 5. Further analysis

### 5.1. Heterogeneity analysis

Age Heterogeneity. Drawing on Zhao's research, the sample was divided into two groups: young respondents (under 45 years old) and middle-aged and elderly respondents (45 years old and above) [61]. The results in Table 10 indicate that, among younger respondents, poorer mental health is driven primarily by the digital access and digital use divides. In sharp contrast, for middle-aged and elderly villagers the digital use divide alone exerts the most pronounced negative effect. This age-pattern likely reflects the life-course realities of rural China: older adults need a broader repertoire of online skills—job search, mobile banking, government-service portals—to secure or retain off-farm employment, so deficiencies in use directly curtail their labour-market opportunities and, in turn, depress psychological well-being. Younger villagers, while more digitally native, still face fundamental barriers to entry (device affordability, village-level fibre availability); once on-line, they demand advanced skills for study, social media and e-commerce. Hence inadequate access and low usage intensity reduce their information-gathering efficiency and generate the observed mental-health penalty.

Gender Heterogeneity. Following Li's (2025) methodology, all respondents were divided into two groups by gender for in-depth analysis [55]. The results show that men's mental health is impaired mainly by the digital use and digital utility divides, whereas women are adversely affected across all three layers. Along both the access and use dimensions, the negative mental-health effect is significantly larger for women; along the utility dimension, the effect is larger for men. These patterns plausibly reflect regional disparities in economic development and gendered educational expectations. Women, who on average face lower access to and fewer opportunities for internet use, tend to focus on its benefits and are less aware of potential downsides; hence each additional barrier to connectivity or usage translates into a sharper psychological penalty. Men, enjoying greater access and usage experience, are more attuned to the risks and limitations of online activity, so the perceived (low) utility of the internet exerts a stronger negative influence on their well-being.

**Table 10. Heterogeneity analysis.**

| Item | Age < 45 | | | Age ≥ 45 | | |
|---|---|---|---|---|---|---|
| Digital Divide | | | | | | |
| Digital Access Divide | −0.174** (−2.05) | | | −0.0910* (−1.79) | | |
| Digital Use Divide | | −0.0937*** (−3.47) | | | −0.143*** (−6.25) | |
| Digital Utility Divide | | | −0.119*** (−5.45) | | | −0.109*** (−6.40) |
| N | 4843 | 4492 | 4492 | 6271 | 2323 | 2323 |
| **Item** | **Male** | | | **Female** | | |
| Digital Divide | | | | | | |
| Digital Access Divide | −0.0925 (−1.55) | | | −0.129** (−2.20) | | |
| Digital Use Divide | | −0.130*** (−4.25) | | | −0.161*** (−4.62) | |
| Digital Utility Divide | | | −0.109*** (−6.40) | | | −0.0746*** (−2.98) |
| N | 5672 | 3560 | 3560 | 5442 | 3255 | 3255 |
| **Item** | **Below associate degree** | | | **an associate degree or higher** | | |
| Digital Divide | | | | | | |
| Digital Access Divide | −0.115*** (−2.70) | | | 0.291 (0.78) | | |
| Digital Use Divide | | −0.154*** (−4.04) | | | −0.0444 (−0.65) | |
| Digital Utility Divide | | | −0.0720*** (−2.73) | | | −0.130** (−1.97) |
| N | 10406 | 6126 | 6126 | 708 | 689 | 689 |
| **Item** | **Eastern region** | | | **Non−eastern region** | | |
| Digital Divide | | | | | | |
| Digital Access Divide | −0.145** (−2.20) | | | −0.0827 (−1.53) | | |
| Digital Use Divide | | −0.191*** (−5.17) | | | −0.116*** (−3.95) | |
| Digital Utility Divide | | | −0.105*** (−4.14) | | | −0.111*** (−4.89) |
| N | 4356 | 2639 | 2639 | 6758 | 4176 | 4176 |

Notes: ① *, **, and *** denote significance at the 10%, 5%, and 1% levels, respectively.

② t-statistics are in parentheses.

③ Individual and time fixed effects are controlled for.

④ Control variables have been accounted for.

Educational Heterogeneity. Referencing Lu's research, the sample was divided into two groups based on educational attainment: "junior college or higher" (higher education) and "below junior college" (lower education) [9]. The mental-health penalty of the digital divide is concentrated among the low-educated, for whom access and use divides reduce information retrieval efficiency and job access. Among the highly educated, the utility divide predominates, constraining perceived advancement opportunities and thereby generating sharper psychological losses.

Regional Heterogeneity. Following Yang's research, respondents were divided into two groups by province: eastern region (residing in eastern China) and non-eastern region (central and western China) [8], with results presented in Table 10. Empirical findings reveal marked regional heterogeneity in the relationship between the digital divide and rural residents' mental health. Specifically, the digital access divide exerts a statistically significant negative effect on rural residents in eastern China (coefficient equals negative 0.145, p value less than 0.05). This result is consistent with the region's advanced economic development and high internet penetration, conditions that render connectivity indispensable for schooling, employment, and daily life; exclusion therefore constitutes a barrier to opportunity and, consequently, a threat to psychological well-being. In non-eastern regions—where internet infrastructure is less developed and online services are not yet integral to every domain—the access divide does not significantly influence mental health. Nevertheless, among non-eastern residents who are connected, both the digital use divide and the digital utility divide continue to impede work efficiency and life convenience, producing inferior mental-health outcomes.

Further disaggregation shows that the use and utility divides operate differently across regions. In the east, the estimated coefficient of the digital use divide is negative 0.191 (p value less than 0.001) and that of the digital utility divide is negative 0.105 (p value less than 0.001), indicating that even when infrastructure is available, inability to exploit digital resources and low perceived importance of the internet both depress psychological states. In non-eastern areas the corresponding coefficients are negative 0.116 and negative 0.111 (both p values less than 0.001), demonstrating that these divides are equally harmful. Lower average schooling and slower economic growth in non-eastern regions likely magnify the damage, because residents possess fewer skills with which to convert digital access into tangible benefits.

## 5.2. Mechanism testing

In this subsection, drawing on the mechanism testing methods proposed in prior studies, we examine the mechanisms through which the digital divide affects the mental health of rural residents in China [53,57]. Specifically, we analyze the mediating roles of Self-Assessed Social Class Perception, Self-Assessed Economic Status, and Self-Assessed Sense of Fairness. The detailed results are presented in Table 11.

Panel A of Table 11 examines whether self-assessed social-class perception mediates the effect of the digital divide on the mental health of rural Chinese residents. Column (1) shows that the digital access divide exerts a significant negative effect on mental health (coefficient = −0.110). Yet column (2) indicates that the access divide does not significantly predict self-assessed social class; hence, no mediating role is detected. Turning to the digital use divide, column (4) reports a significant negative coefficient of −0.143 on mental health. Column (5) reveals that the use divide also significantly lowers self-assessed social class. After controlling for the latter in column (6), the coefficient on mental health remains significant and only marginally attenuates to −0.142, implying that the use divide harms mental health partly by depressing perceived social status.

Finally, column (7) documents that the digital utility divide significantly reduces mental health (coefficient = −0.109, p < 0.01). Column (8) shows the same divide significantly lowers self-assessed social class. Introducing the mediator in column (9) leaves the negative effect on mental health virtually intact (coefficient = −0.107), confirming that the utility divide operates, at least partially, through diminished perceptions of social standing.In summary, the digital usage and utility divides influence mental health by altering Self-Assessed Social Class Perception, while the digital access divide does not. Thus, Hypothesis H2 is partially supported.

Panel B of Table 11 examines whether Self-Assessed Economic Status mediates the relationship between the digital divide and mental health. Columns (7) to (9) show that the digital utility divide has a significantly negative effect on mental health. Specifically, column (7) indicates a coefficient of −0.109, while columns (8) and (9) demonstrate that the digital utility divide significantly reduces Self-Assessed Economic Status. When controlling for Self-Assessed Economic Status, the coefficient of the digital utility divide on mental health remains robust, decreasing slightly to −0.107. This suggests that the digital utility divide negatively affects mental health by reducing Self-Assessed Economic Status. In contrast, the digital access and usage divides do not exhibit this mediating effect. Therefore, Hypothesis H3 is partially supported.

Panel C of Table 11 explores whether Self-Assessed Sense of Fairness mediates the relationship between the digital divide and mental health. Columns (1) to (3) show that the digital access divide negatively affects mental health, with a coefficient of −0.110 in column (1). Columns (2) and (3) further indicate that the digital access divide significantly reduces Self-Assessed Sense of Fairness. When controlling for Self-Assessed Sense of Fairness in column (3), the coefficient of the digital access divide on mental health remains negative and significant, decreasing slightly to −0.108. This demonstrates that the digital access divide negatively impacts mental health by reducing Self-Assessed Sense of Fairness. The digital usage and utility divides, however, do not exhibit this mediating effect.

In sum, the digital access divide undermines the mental health of rural Chinese residents primarily by eroding their perceived sense of fairness; the digital use divide does so by diminishing their self-assessed social-class position; and the

**Table 11. Mechanism Test.**

**A Self-Assessed Social Class Perception**

| Var | Mental Health Levels | Class | Mental Health Levels | Mental Health Levels | Class | Mental Health Levels | Mental Health Levels | Class | Mental Health Levels |
|---|---|---|---|---|---|---|---|---|---|
| Item | (1) | (2) | (3) | (4) | (5) | (6) | (7) | (8) | (9) |
| Digital Access Divide | −0.110*** (−2.63) | −0.000930 (−0.02) | −0.110*** (−2.63) | | | | | | |
| Digital Use Divide | | | | −0.143*** (−6.25) | −0.0529** (−2.24) | −0.142*** (−6.18) | | | |
| Digital Utility Divide | | | | | | | −0.109*** (−6.40) | −0.0814*** (−4.68) | −0.107*** (−6.27) |
| Class | NO | NO | YES | NO | NO | YES | NO | NO | YES |
| N | 11114 | 11114 | 11114 | 6815 | 6815 | 6815 | 6815 | 6815 | 6815 |

**B Self-Assessed Economic Status**

| Var | Mental Health Levels | Condition | Mental Health Levels | Mental Health Levels | Condition | Mental Health Levels | Mental Health Levels | Condition | Mental Health Levels |
|---|---|---|---|---|---|---|---|---|---|
| Item | (1) | (2) | (3) | (4) | (5) | (6) | (7) | (8) | (9) |
| Digital Access Divide | −0.110*** (−2.63) | 0.0440 (0.85) | −0.111*** (−2.65) | | | | | | |
| Digital Use Divide | | | | −0.143*** (−6.25) | −0.0324 (−1.27) | −0.143*** (−6.22) | | | |
| Digital Utility Divide | | | | | | | −0.109*** (−6.40) | −0.099*** (−5.30) | −0.107*** (−6.29) |
| condition | NO | NO | YES | NO | NO | YES | NO | NO | YES |
| N | 11114 | 11114 | 11114 | 6815 | 6815 | 6815 | 6815 | 6815 | 6815 |

**C Self-Assessed Sense of Fairness**

| Var | Mental Health Levels | Fair | Mental Health Levels | Mental Health Levels | Fair | Mental Health Levels | Mental Health Levels | Fair | Mental Health Levels |
|---|---|---|---|---|---|---|---|---|---|
| Item | (1) | (2) | (3) | (4) | (5) | (6) | (7) | (8) | (9) |
| Digital Access Divide | −0.110*** (−2.63) | −0.365*** (−3.20) | −0.108*** (−2.58) | | | | | | |
| Digital Use Divide | | | | −0.143*** (−6.25) | −0.0521 (−0.89) | −0.143*** (−6.23) | | | |
| Digital Utility Divide | | | | | | | −0.109*** (−6.40) | −0.0671 (−1.56) | −0.108*** (−6.38) |
| Fair | NO | NO | YES | NO | NO | YES | NO | NO | YES |
| N | 11114 | 11114 | 11114 | 6815 | 6815 | 6815 | 6815 | 6815 | 6815 |

Notes: ① *, **, and *** denote significance at the 10%, 5%, and 1% levels, respectively.

② t-statistics are in parentheses.

③ Individual and time fixed effects are controlled for.

④ Control variables have been accounted for.

digital utility divide exerts its adverse effect through both lowered self-assessed social class and reduced self-assessed economic status.

Several mechanisms underlie these patterns.First, insufficient connectivity curtails access to information and online social participation, fostering doubts about distributive justice and, consequently, psychological distress.

Second, as labour markets increasingly reward digital competencies, the inability to use the internet productively restricts employment opportunities, prompting negative evaluations of one's own social standing. Third, perceiving the internet as unimportant intensifies feelings of social exclusion: the widening status gap implied by the utility divide breeds

pessimism about one's relative position, while diminished capacity to engage in e-commerce or digital entrepreneurship constrains income growth and lowers perceived economic status, ultimately damaging mental well-being [62,63].

## 6. Discussion

The continuous advancement and popularization of internet technology have transformed the digital divide from a primary "digital access divide" into second-level "digital use divide" and third-level "digital utility divide." Using data from CFPS2020 and CFPS2022, this study deeply explores the impact of the digital divide on the mental health of rural Chinese residents. Existing research on this topic is relatively limited, with most studies focusing on the effects of the digital access and use divides, while the impact of the digital utility divide on rural residents' mental health remains under-researched.

The findings reveal that when rural residents lack internet access, exhibit low internet usage frequency, or perceive low importance of the internet, their mental health declines. Sub-dimension analysis shows that the absence or improper use of online learning skills, online shopping skills, and short video skills are associated with reduced mental health. Additionally, the negative impact of internet use on maintaining contact with family and friends also adversely affects rural residents' mental health. Furthermore, the influence of the digital divide on mental health is moderated by factors such as age, gender, education level, and regional location. Specifically:The digital access divide impairs mental health by reducing residents' Self-Assessed Sense of Fairness. The digital use divide lowers mental health through decreasing Self-Assessed Social Class Perception. The digital utility divide affects mental health by reducing both Self-Assessed Social Class Perception and Self-Assessed Economic Status.

Relative to the existing literature, the present study makes two principal contributions. First, we extend the conventional two-tier conceptualisation of the digital divide (access versus use) to a three-tier framework by incorporating the digital utility divide. Prior work has overwhelmingly operationalised the divide as a binary of "access–use", thereby neglecting the deeper implications of perceived internet utility. Our findings not only furnish robust evidence for the existence of the utility divide, but also dissect its nuanced psychological consequences for rural residents. This theoretical extension furnishes an analytical template for future scholarship. Second, we offer a focused examination of rural populations in a developing-country context, thereby redressing the paucity of mental-health research on this demographic. Rather than restricting attention to canonical "farmers", we broaden the analytical lens to encompass all agricultural and non-agricultural workers resident in rural areas—an innovation that carries direct implications for place-based policy design. Additionally, by disag-gregating the three divides and tracing their differential pathways—via self-assessed social class, self-assessed economic status, and perceived fairness—we provide a level of mechanistic detail that is seldom available in extant studies and that is indispensable for targeted intervention. Notwithstanding these strengths, full resolution of reverse causality remains elusive. Rural Chinese residents confront objectively constrained social channels and cognitive repertoires relative to their urban counterparts. Following Sattar and Øland, two feedback loops are plausible. First, depressive symptomatology may precipitate social withdrawal, eroding the perceived need to go online; low self-efficacy leads individuals to forgo the internet as a means of social or informational expansion, manifesting as access and use divides [64]. Second, depression can impair technology-learning capacity, causing already-connected individuals to abandon skill acquisition and to devalue internet utility—an endogenous process that surfaces as a utility divide. Future work should employ randomised or cross-over designs to neutralise these reciprocal influences [65]. Nevertheless, the battery of endogeneity tests deployed here—IV estimation, Hausman specification tests, and partial-$R^2$ metrics—together with an extensive suite of robustness checks, sustains the credibility and policy relevance of the core findings.

## 7. Implications

### 7.1. Policy implications

The results of this study hold significant value for the advancement of the Digital China strategy and public health interventions. The Digital China strategy aims to achieve high penetration and high-quality development of the

Internet. The heterogeneity analysis in this study demonstrates the impact of the digital divide on the mental health of different groups of rural residents, which will provide references for the precise advancement of the Digital China strategy. For example, the focus of high penetration development should be shifted to the central and western regions or middle-aged and elderly groups. The focus of high-quality development should be on rural areas in the eastern region or rural residents with higher education levels, thereby reducing the waste of efficiency. As an important part of public health interventions, the response to mental health issues is also of great significance for its optimization. The results show the differences in the mechanisms through which different levels of the digital divide affect the mental health of rural residents, which will provide fundamental intervention pathways for public health interventions to alleviate the mental health problems of digital vulnerable groups such as rural residents, thereby enhancing the effectiveness of interventions.

The results of this study also provide certain value enlightenment for grassroots policy-making. The results show the differences in the impact of the digital access divide, digital use divide and digital utility divide on the mental health of different groups of rural residents, which is conducive to the local governments to propose specific and targeted measures for different groups of rural residents.

Age-Specific Interventions. For rural youth,the strategy should involve expanding mobile internet coverage in remote areas and integrating digital skills training into pre-employment services. This includes incorporating skills like online marketing, remote work software, and short-video creation into training programs, and establishing a digital skills certification system to link skill acquisition with better job opportunities. This approach can enhance digital engagement and mitigate the psychological harm caused by access and use divides. For middle-aged and elderly rural residents, emphasizing the internet's role in maintaining intergenerational connections and reducing caregiving burdens can alleviate their resistance to digital technology and the psychological impact of the use divide.

Gender-Specific Interventions.For women, efforts should focus on breaking down barriers to internet access and use, providing sustained digital tool training to reduce the psychological impact of these divides. For men, the emphasis should be on addressing the adverse effects of potential internet risks, enhancing their recognition of the internet's value and safety, and thereby mitigating the negative impact of the digital utility divide.

Education-Specific Interventions.For rural residents with lower education levels, promoting basic digital skills training can help overcome cognitive barriers and reduce the psychological impact of the digital divide. For those with higher education, advanced digital skills training, such as digital marketing analysis, should be promoted to create more high-quality job opportunities and prevent skill-demand mismatches.

Region-Specific Interventions.In eastern rural areas, improving internet speed and stability while increasing the frequency of digital applications in daily life is essential. In central and western rural regions, priority should be given to ensuring basic internet access and optimizing base station construction and signal strength, along with long-term plans for digital skills enhancement. These measures can effectively reduce the adverse effects of the digital divide.

## 7.2. Academic implications

First, academic research should be closely integrated with societal realities, focusing not only on theoretical inquiry but also on practical relevance when addressing research questions. When studying factors affecting the mental health of rural residents, researchers should prioritize the representativeness and generalizability of sample selection. When examining the impacts of the digital divide, dividing it into hierarchical dimensions using reasonable criteria can yield more detailed and precise conclusions, helping to reduce sampling errors and ensure the robustness of findings [6].

Additionally, as internet technology continues to develop, the scope and content of the digital divide have evolved into a complex and diverse field [14]. Academic research must keep pace with these changes by incorporating relevant influencing factors and optimizing existing research frameworks, which is crucial for conducting comprehensive discussions.

Finally, there is a need for thorough research on how the digital divide affects the mental health of rural Chinese residents, including investigations into differential impacts across different age groups, regions, and occupations. Exploring potential mediating mechanisms will provide support for problem-solving and dynamically understanding the issue.

## 8. Conclusions

This study examines the impact of the digital divide on the mental health of rural Chinese residents. First, it provides operational definitions for three dimensions of the digital divide: the "digital access divide," "digital use divide," and "digital utility divide," using this logical framework to elaborate on the current status of China's digital divide. Second, the findings reveal that regardless of the dimension—access, use, or utility—the digital divide is associated with lower mental health among rural residents, with effects moderated by factors such as age, gender, education, and region. Third, different digital divide dimensions operate through distinct mediating mechanisms:The digital access divide impairs mental health by reducing rural residents' Self-Assessed Sense of Fairness.The digital use divide negatively affects mental health by decreasing Self-Assessed Social Class Perception.The digital utility divide lowers mental health through impacts on both Self-Assessed Social Class Perception and Self-Assessed Economic Status.

## 9. Research limitations and future directions

This study also has some limitations and shortcomings. First, the study scope is limited to rural residents in China. While this focus strengthens the robustness of conclusions for this specific group, it limits the generalizability of the findings. Expanding the scope could enhance generalizability but may introduce greater uncertainty. Second, although this study uses an instrumental variable (IV) to address endogeneity issues such as omitted variables, measurement errors, and bidirectional causality, it cannot completely avoid the influence of endogeneity. Third, this study uses data from the China Family Panel Studies (CFPS) in 2020 and 2022 to investigate the impact of the digital divide on the mental health of rural residents in the post-pandemic era, but it cannot deeply identify the dynamic impact and mechanism of the digital divide on rural residents' mental health over a longer period.

To address these limitations, future research should employ data with longer time spans to investigate the relationship between the digital divide and rural mental health. Additionally, more comprehensive and rigorous measurement methods should be designed, more variables should be included, and more reasonable instrumental variables should be tried to mitigate the impact of endogeneity. Moreover, further exploration of potential mediating mechanisms between the digital divide and rural mental health is warranted.

## Supporting information

**S1 Data. Raw dataset related to the digital access divide.**
(XLSX)

**S2 Data. Raw datasets related to the digital use divide and digital utility divide.**
(XLSX)

## Author contributions

**Conceptualization:** Yunhui Ai.

**Data curation:** Yi Ding.

**Funding acquisition:** Yunhui Ai.

**Methodology:** Yi Ding.

**Software:** Yi Ding.

**Supervision:** Yi Ding.

**Visualization:** Yi Ding.

**Writing – original draft:** Yi Ding.

**Writing – review & editing:** Yi Ding.

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
