## [Decision Letter · Decision Letter 0]

5 Sep 2025

Dear Dr. Ai,

Thank you for submitting your manuscript to PLOS ONE. After careful consideration, we feel that it has merit but does not fully meet PLOS ONE’s publication criteria as it currently stands. Therefore, we invite you to submit a revised version of the manuscript that addresses the points raised during the review process.

We look forward to receiving your revised manuscript.

Kind regards,

RAMYA KUNDAYI RAVI

Academic Editor

PLOS ONE

Journal Requirements:

3. Please ensure that you refer to Figure 1 in your text as, if accepted, production will need this reference to link the reader to the figure.

4. We are unable to open your Supporting Information file [data1.dta, data2.dta, do.do]. Please kindly revise as necessary and re-upload.

Reviewers' comments:

Reviewer's Responses to Questions

**Comments to the Author**

1. Is the manuscript technically sound, and do the data support the conclusions?

Reviewer #1: Yes

Reviewer #2: Yes

2. Has the statistical analysis been performed appropriately and rigorously?

Reviewer #1: Yes

Reviewer #2: Yes

3. Have the authors made all data underlying the findings in their manuscript fully available?

Reviewer #1: Yes

Reviewer #2: Yes

4. Is the manuscript presented in an intelligible fashion and written in standard English?

Reviewer #1: Yes

Reviewer #2: Yes

Reviewer #1: The manuscript is strong, addressing a highly relevant topic with a robust methodology and thorough analyses; the three-level digital divide concept is particularly valuable. However, few areas need revision for clarity.

In the introduction, the manuscript defines its sample as "rural residents" but heavily emphasizes "farmers”. If "rural residents" is the broader focus, the introduction needs to acknowledge the diversity within this group and include literature review about the remaining group of the population.

Within the methods section, an explicit statement detailing the ethical approval for the original CFPS data collection and how participant consent was obtained must be included, even for secondary data use.

Also in the methods, since new composite variables (e.g., mental health, digital use, digital utility) were created using PCA, psychometric evidence for these in the sample must be provided.

Regarding variable detailing in the methods, when explaining variables in the text, it is not necessary to list every single item if the full details are already provided in a table. A concise explanation with a few key examples is sufficient in the paragraph.

Finally, the paper lacks a dedicated "Discussion" section. A standalone discussion is essential for synthesizing findings, comparing them with existing literature, and exploring theoretical and policy implications in depth.

Reviewer #2: Strengths

Clear title and abstract reflecting the scope of the research.

Well-structured introduction with a detailed background and clear hypotheses.

Use of a large, nationally representative dataset, enhancing the generalizability of findings.

Robust statistical analysis including fixed effects, robustness checks, IV estimation, and heterogeneity analyses.

Areas for Improvement

Methodological transparency

Provide more detail on the PCA used to derive the mental health, digital use, and digital utility indices, including loadings, explained variance, and reliability tests.

Clarify handling of missing data and whether sample weights were applied.

Instrumental variable justification

The chosen IV (mean digital divide of the same-age villagers) requires a stronger theoretical and empirical justification, including more detailed diagnostics for strength and validity.

Ethics statement

The ethics section currently states “N/A,” which is not acceptable. Explicitly describe the IRB approvals from CFPS, informed consent procedures, and compliance with data-use agreements.

Results clarity

Include 95% confidence intervals and standardized coefficients to allow better interpretation of effect sizes.

Summarize key findings more concisely in the text to complement complex tables.

Tables and figures

Clean up formatting issues, especially in Table 1 where items are duplicated. Ensure Figure 1 is clear and labeled.

Language and style

Edit for grammar and clarity. For example, “55st statistical report” should be “55th statistical report.”

Discussion and conclusions

Expand on practical implications of the findings and discuss reverse causality more critically.

Offer more actionable policy recommendations tailored to different demographic groups.

**Do you want your identity to be public for this peer review?** For information about this choice, including consent withdrawal, please see our Privacy Policy

Reviewer #1: No

Reviewer #2: **Yes:** Tirtjaraj Acharya

---

## [Author Response · Author response to Decision Letter 1]

19 Sep 2025

Dear Editors and Reviewers,

I hope this message finds you well.

I have thoroughly reviewed the comments and suggestions provided by the reviewers and the editors. I sincerely appreciate the time and effort you have dedicated to reviewing our manuscript. Your feedback has been invaluable in helping us improve our research.

In response to the comments, I have prepared a detailed rebuttal letter titled "Respond to the reviewers," which addresses each point raised by the reviewers. In this document, I have provided comprehensive responses, including specific revisions made to the manuscript, explanations for our methodology, and additional analyses where necessary. The rebuttal letter is attached to this submission for your reference.

I believe that the revisions and responses in the rebuttal letter have significantly enhanced the quality and clarity of our manuscript. I am confident that these changes meet the journal's standards and address the concerns raised during the review process.

Please let me know if there is any further information I can provide or if there are any additional questions or concerns that need to be addressed.

Thank you once again for your careful consideration and support.

Best regards,

Yi Ding

---

## [Decision Letter · Decision Letter 1]

5 Jan 2026

The Impact of the Three-Level Digital Divide on the Mental Health of Rural Residents: A Study from China

PONE-D-25-20224R1

Dear Dr.

We’re pleased to inform you that your manuscript has been judged scientifically suitable for publication and will be formally accepted for publication once it meets all outstanding technical requirements.

Kind regards,

RAMYA KUNDAYI RAVI

Academic Editor

PLOS One

---

## [Editor Report · Acceptance letter]

PONE-D-25-20224R1

PLOS One

Dear Dr. Ai,

I'm pleased to inform you that your manuscript has been deemed suitable for publication in PLOS One. Congratulations! Your manuscript is now being handed over to our production team.

Kind regards,

on behalf of

Dr. RAMYA KUNDAYI RAVI

Academic Editor

PLOS One